# Defect Detection Method for CFRP Based on Line Laser Thermography

**DOI:** 10.3390/mi13040612

**Published:** 2022-04-13

**Authors:** Quan Wang, Zhijie Zhang, Wuliang Yin, Haoze Chen, Yushan Liu

**Affiliations:** 1School of Instrument and Electronics, North University of China, Taiyuan 030051, China; w1341042446@163.com (Q.W.); wuliang.yin@manchester.ac.uk (W.Y.); 15930663972@163.com (H.C.); liuyushan_nuc@163.com (Y.L.); 2Key Laboratory of Instrumentation Science & Dynamic Measurement, North University of China, Ministry of Education, Taiyuan 030051, China; 3School of Electrical and Electronic Engineering, University of Manchester, Manchester M13 9PL, UK

**Keywords:** NDT, carbon-fiber-reinforced polymer, defect location, finite element method

## Abstract

A continuous line laser scanning inspection technique for tracing load-bearing structures was developed and applied to defect detection of unidirectional carbon-fiber-reinforced polymers for aero engines. The heat transfer model of the material was analyzed using the finite element software COMSOL. Meanwhile, a laser platform was built and an image algorithm was used to verify the feasibility of the method. The potential of this technique for detecting defects and providing information on the location of defects in carbon fiber composites was analyzed. Results indicate line laser thermal imaging can successfully determine the size, location, and crack angle of surface damage with extremely high accuracy. The positioning accuracy error for impact and fracture defects is less than 20%, and the detection rate can reach 100% if the defect is in the special position of just leaving the heating area. The angle detection of fracture cracks can be accurate within 10°.

## 1. Introduction

The tremendous advances in materials manufacturing technology have led to the use of carbon-fiber-reinforced polymers with excellent properties for aerospace, vehicles, ships, construction, and many other fields [1,2]. As far as engineering applications are concerned, the complex and variable environments in which carbon fiber is used have led to a decrease in the overall performance of carbon fiber [3,4,5,6]. There is a great possibility of damage, which cannot meet the design and use requirements. Therefore, it is necessary to evaluate the safety and reliability of materials and components through a series of inspections at each stage of production and maintenance. There are many means that are used quite often by engineers and researchers. Ultrasound [7], radiation, eddy current [8], magnetic particle [9], and infiltration [10] methods, along with the method we use, infrared evaluation [11], are just a few of the vast array of inspection modalities available to engineers and academics. Every detection technique has its own area of application, with advantages and disadvantages, and infrared thermography is no exception [12,13,14,15]. It uses an external thermal source (such as laser) to change the surface temperature distributions of materials, thereby forming 3D heat flow conduction inside the solid [16,17,18]. T800H carbon fiber composite laminate used can replace the metal (or alloy) load-bearing structure of an aero engine. Applications have shown that this composite material can withstand certain temperature changes. The main reason why a laser can be used as an external heat source is that it has advantages that other heat sources, such as halogen lamps and flash lamps, cannot provide. It can output high power and is controllable, the temperature of the heating area is uniform, and the energy scattering in the air is small. These all demonstrate the great potential and advantages of lasers in industrial inspections or processing. A thermal imager is utilized to monitor the temperature response of the surface. The differential distribution of temperature can identify healthy and damaged areas. In other words, optical radiation measurement of the damage is based on the principle that the damage causes irregular heat flow nearby [3].

Generally, infrared images captured by thermal imaging cameras have low contrast between the target and the background, resulting in blurring of the physical characteristics of the defect being inspected. It is also difficult to directly judge the existence of damage from a thermal sequence image. Most researchers are innovating detection methods to achieve the effect of enhancing contrast. Wang et. al. [19] presented a transverse heat flow suppression (THFS) technique which demonstrated that enhanced TWRT can realize the effective detection of defects (90% detection probability) with a diameter depth ratio of 5.06 under a 95% confidence level. Shi et. al. [20] presented a comparative experiment between BC-TWI and lock-in thermography (LIT), achieving higher contrast and SNR than the LIT phase image. Y et. al. [21] used a vibrothermography to detect and evaluate low-energy impact damage. Wang et. al. [22] investigated differential spread laser infrared thermography (DSLIT), and a defect at a depth of 2.5 mm was detected successfully, with the width-to-depth ratio of up to 2.4. Fei et. al. [23] proposed a FFT phase feature method and provided a rapid NDT&E and 3D tomography method for CFRP with subsurface defects under the low-energy excitation condition. N. Montinaro [24] used continuous laser scanning of materials’ surfaces to detect interlaminar debonding defects in aerospace-grade fiber laminates and briefly characterized the locations and sizes of surface defects.

The purpose of this study was to investigate the detectability and location information of surface damage in unidirectional carbon fiber laminates under continuous line laser scanning thermography. The controllable laser was applied to carbon fiber defect detection, and the temperature change on the material’s surface was observed by a thermal imager. Firstly, a heat conduction model based on the finite element method COMSOL is proposed for this problem. With it, we simulated a line laser scanning experiment with a specimen. It provided a range of parameter choices for subsequent experiments. We give an experimental path based on this idea. The infrared images were inspected by image processing, the defects were successfully identified, and localization analysis was carried out. The results show that the method can detect the specific location information of defects.

## 2. Principle Analysis and Model Structure Introduction

In this section, we will explain in detail the material parameters used in finite element analysis. These parameters were used to study the surface temperature change of unidirectional carbon fiber composites under continuous line laser scanning.

The geometric parameters of the carbon fiber used in the paper are shown in Figure 1. This composite type is T800H, and its properties were similar to those produced by TORAY. Its laying direction was 0°, it had a total of 14 layers, and the overall size was 100 mm × 167 mm × 2.4 mm. It consisted of defects of different sizes. It is mainly used to replace metal load-bearing parts in aero engines to reduce weight. Defects were manually cut to simulate various conventional damages that may occur in an actual production process. For example, impact damage was simulated with discs of different sizes and depths (Φ3 and Φ5 mm), and tensile fractures (1 mm × 10 mm) were simulated with different open cracks. The simulated damage thickness gradually increased from 0.5 to 2 mm at equal intervals of 0.5 mm. According to the microstructural characteristics of the carbon fiber sample, it is known that the diameter of each single fiber is very small (5 μm), so the fiber sample can be equivalent to a cuboid with the same properties. We know that the thermal anisotropy of a fiber is determined by the thermal conductivity of the material along the axis. We checked the literature and material property table, and selected the thermophysical parameters shown in Table 1 for finite element simulation analysis [25]. In terms of heat conduction, the thermal conductivity in the axial direction is much higher than the thermal conductivity in the other directions. Each defect was filled with air, and it is not described in this paper. Defect types are specified with numbers 1–5, which are 3 mm, 5 mm, 0°, 90°, and 45°, respectively.

In this work, the simulated line laser length was 25 mm. Figure 2 shows the laser moving mode, i.e., vertical movement. It is easy to understand that the motion is related to the position change in 2D space. Figure 2a shows the coordinate change of the laser center. Figure 2b shows the numerical values 0 and 1 to simulate the switching state of the laser. The heating time used in the experiment was 0–20 ms, and cooling was 20–25 ms. These states are exploited to accurately control the operation of the simulated laser, and the temperature distribution on the surface of the material is observed and recorded. The incident heat flux function describes the frontal (heat source boundary) input of the carbon fiber composite sample, as shown below.
(1)$$q^{in}\left(x,0,z,\tau \right)=\frac{2p}{\pi R^{2}}\times \exp(-\frac{2r^{2}}{R^{2}})\times em\times pw\left(t\right)$$
(2)$$r=\sqrt{\frac{\left(x-x_{0}{}^{2}\right)}{0.1}+\frac{\left(z-z_{0}{}^{2}\right)}{10}}$$
where $q^{in}\left(x,0,z,\tau \right)$ is the heat flux function of the surface heat source, p is the laser modulated power (14 W), *R* is the diameter of the laser spot (4 mm), and em is the surface emissivity of the material. pw(t) is the laser running time, which indicates its status. The coordinate $\left(x_{0},z_{0}\right)$ is the start point when the laser focus moves in a specified way (as seen in Figure 2). We measured the appearance of the sample and then designed the laser shape to ensure the integrity and accuracy of the sample scan. Therefore, the purpose of Equation (2) is to convert a spot distributed over a concentric circle into an elliptical laser beam that approximates a line segment.

In the theory of heat transfer, when considering the thermal anisotropy of materials, the temperature distribution *T*(*x*, *y*, *z*, *t*) of solid materials should satisfy the following partial differential equation of heat conduction [26]:(3)$$\begin{array}{l} \rho \left(x,y,z\right)\cdot c_{p}\left(x,y,z\right)\frac{\partial T(x,y,z,t)}{\partial t}-[\frac{\partial }{\partial x}\left(k_{xx}\left(x,y,z\right)\frac{\partial T\left(x,y,z,t\right)}{\partial x}\right)+\\ +\frac{\partial }{\partial y}\left(k_{yy}\left(x,y,z\right)\frac{\partial T\left(x,y,z,t\right)}{\partial y}\right)+\frac{\partial }{\partial z}\left(k_{zz}\left(x,y,z\right)\frac{\partial T\left(x,y,z,t\right)}{\partial z}\right)]=0 \end{array}$$
where ρ(x,y,z) is the density of solid material and $k_{d}(x,y,z)$ is thermal conductivity (*d* = *xx*, *yy*, *zz*).

The experiment was set up and analyzed with the described parameters. When observing the abnormal distribution of the temperature field, it was found that only one-dimensional coordinates could be determined. In other words, the ordinate of the current laser center was damaged, and the abscissa was roughly within half of the length (±12.5 mm). Therefore, a single scan did not accurately reflect the coordinates of the abnormal location. Different scanning methods were used to address this phenomenon; that is, the scanning direction was perpendicular to the mentioned motion pattern. We used a solid heat transfer module to simulate the thermal response behavior under pulsed heat source motion and fully considered the boundary conditions, namely, Equations (1) and (2).

## 3. Experiment

In this part, the principle of line laser scanning experiment will be introduced in detail. The laser pulse heat wave experiment is shown in Figure 3.

It is obvious that the entire experimental system consists of an external excitation source, optical system, sample, and signal receiver. The optical system consists of two lenses, namely, plano-convex lens and plano-convex cylindrical lens, which mainly change the laser shape. Figure 3b shows the geometry of the line laser in the experiment. A high intensity, high energy concentration continuous heat source acts on the composite material. The synchronized control platform changes the position of the sample in space according to a defined speed or course. The temperature changes of the material are recorded with a thermal imager. We believe that differences in physical properties (e.g., thermal conductivity) between damaged and healthy regions cause anomalous distribution of the surface temperature field [12,27,28]. The corresponding infrared radiation intensity will also be different.

For example, when a material exhibits any defects in its structure, its material thickness, thermal conductivity, and density can change, affecting heat transfer within the material. Another consideration, without a doubt, when conducting thermal nondestructive testing is the thermal diffusivity of the material to be tested [29]. In the thermal nondestructive investigation, the factors mentioned in the appeal will cause the temperature change of the material [3,30]. These can be realized from the following equation:(4)$$\alpha =\frac{k}{\rho c_{p}}$$
(5)$$T=\frac{Q}{e\sqrt{\pi t}}$$
(6)$$e=\sqrt{k\rho c_{p}}$$
where α is the thermal diffusivity (m^2^·s^−1^), *k* is the thermal conductivity (W·m^−1^·K^−1^), ρ is the density (kg·m^−3^), $c_{p}$ is the specific heat capacity (J·kg^−1^·K^−1^), *T* is the Material temperature, *Q* is the input energy (J), and *e* is the thermal effusivity (W·s^1/2^·m^−2^·K^−1^).

The infrared thermal experiments were performed using the range of parameters provided by the simulation, setting the laser power to 13 W. The maximum temperature of the material’s surface was 80 °C. During the heating of the carbon fiber material, there were no obvious burn marks on the surface of the sample. This indicates that there was no significant physical change on the surface of the laminate after laser irradiation. Changing the shape of the laser from a 12 mm point laser to a 45 mm line laser increased the scanning efficiency by a factor of three.

## 4. Results and Discussion

To explore the results of using continuous line laser to scan unidirectional carbon fiber damage, the entire experimental analysis process was designed according to Figure 4. In other words, the experiments combined theory and practice to investigate unidirectional carbon fiber damage.

### 4.1. Explanation of Thermal Simulation Results

Next, thermal simulation results are presented to explore the relationship between damage and temperature. Observe the different responses of defect shape and depth to surface temperature. When the laser scans the material, the heat transfer process inside the solid can be roughly divided into the following parts: the total energy injected by the laser, heat loss due to thermal radiation, and thermal absorption at the surface of the material. Since carbon-fiber-reinforced composites are opaque, energy cannot be transmitted through the material during heat transfer, so the heat loss is very small and can be ignored. The energy absorbed per unit area of the surface of the material can be calculated using the following formula:(7)$$E_{suf}=E-E_{l}-E_{d}$$
where $E_{suf}$ is the heat absorbed by the material per unit area, *E* is the total heat injection per unit area of the line laser, *E_l_* is the sum of the heat radiation loss energy of the material and the energy lost by the heat exchange between the material and the air, and *E_d_* is the energy absorbed by the damage area due to the increase in temperature.

Due to the integrity of the material and the high thermal conductivity along the carbon fiber extension, the surface temperature of the material during scanning will characterize its structure. The temperature field of the material is uniformly distributed as the laser scans the healthy area. Typically, the defect is filled with a small amount of gas, which is thermally different from the material itself. When a laser scan encounters such an area, it detects weak thermal conductivity and strong thermal insulation. This will lead to huge abnormal fluctuations in the temperature curve of the damage, showing local “dark spots” on the material’s surface. We mainly studied the temperature change of the material. We know that the defect temperature is approximately equal to the difference between the temperature of the surface and the temperature rise in the damaged area (as shown in Figure 5). As the presence of defects changes the heat transfer properties of the material, some values on the temperature curve will show a sharp rise. Increases or decreases in the parameters in Equation (7) can well explain a temperature change trend.

To illustrate the correlation between temperature and size and depth, different damage models were designed for comparison. The temperature relationship between the line laser and the fracture crack at a certain angle was also explored. The temperature of the damage center was regarded as the actual temperature value. Figure 6a,b shows the temperature response curves of the impact damages of different sizes. It can be clearly seen that the temperature for the same diameter of defect increases exponentially with increasing depth. This temperature buildup will enhance the visual appearance of images in laser scanning thermal imaging. In addition, when the depth is certain, the temperature tends to rise as the defect diameter increases. Figure 7 reveals the temperature response curves of damage in different directions caused by strain. The line laser in operation and these surface damages were at a certain angle in space. It is not difficult to find through the data comparison that the center temperature of the 0° crack was the highest, and its numerical growth trend is also sharper than those of other cracks. By observing the temperature changes of different defects, it can be concluded that the type, size, and depth of damage all affect the temperature change. We can observe from this that impact damage had a relatively high detection rate, because it had the most drastic temperature change, resulting in the best visualization of the image. For cracks, the larger the angle, the smaller the temperature rise around the damage during laser scanning. The scanning direction also has a significant effect on the detection of tensile fractures, so multiple scanning directions should be considered.

Figure 8 plots two different curves for the average temperature and temperature sensitivity of the material. Temperature sensitivity refers to the difference between the highest temperature and the lowest temperature of the material domain. Equation (7) explains in detail the reason for the sharp fluctuations of the curve. It is obvious that the damage has a specific physical size, so the temperature change at the damage point should last for a period of time. According to the formula mentioned, it is known that the energy absorbed by the damage in a short time will be very small. Then the natural phenomenon that comes to mind is that there must be an abnormal temperature peak here, and this value is much larger than the temperature fluctuation value of the healthy area (as shown in Figure 8, from 30 to 40 s). In other words, if it is a narrow mutation point or spike, it can be considered to be caused by the simulated boundary effect and temperature fluctuations in the healthy area, which is an abnormal change that can be ignored. This phenomenon is not considered as an abnormal fluctuation caused by damage interference, but can be considered as a suspicious point. For verification, a line laser or spot laser can be used for fixed-point heating to observe whether the surrounding temperature changes. As the laser continues to scan, the sample will gradually absorb heat, causing the average temperature to gradually increase.

In fact, specific changes that occur at any given time may be of interest. Intuitively, we know that temperature is related to the change in thickness of something under a particular premise. Any significant deviation of temperature anywhere from the surrounding temperature is usually considered to indicate the presence of a defect.

As we can see in Figure 9, the blue circles represent damage centers with certain physical dimensions, and the red asterisks are the coordinate values automatically detected by the algorithm. Using the laser trajectory map and temperature change, the location of the defect was calculated and visualized. Table 2 shows the damage detection results of unidirectional-carbon-fiber-based online laser scanning infrared thermal imaging. They are based on the ratio of the Euclidean distance of the coordinates to the size of the object (referring to half the radius or length). Numbers 1–5 represent different types of defects. In the results in the figure, there is a point that was detected twice; there may have been repeated heating when the line laser scans back and forth. This is also the reason for inaccurate detection. By analyzing the results of the graphs and tables, it can be known that the detection accuracy for simulated delamination disc-shaped defects is higher than the detection accuracy for different tensile fracture damage. It can be seen that the direction and type of the damage impact thermal detection, and the detection error for defects perpendicular to the scanning direction was the largest (number #4). Therefore, it is necessary to perform a complete scan along the horizontal axis and the vertical axis to find cracks in different opening directions. It can be concluded from the coordinate detection results that this method has a certain accuracy for damage location and meets the basic requirements of detection. The laser or sample is made to move at a certain speed, and then the temperature parameter changes on the surface of the material are monitored in real time, and the size and location of the damage are obtained by analysis. This method is completely feasible and can also reflect the characteristics of damage.

### 4.2. Experimental Research of Carbon Fiber Defect Detection

A continuous laser was used to scan and explore unidirectional carbon fiber composites. We studied the thermal response of damage from two aspects: temperature analysis and image processing.

When the laser scans the surface of the material, the thermal equilibrium on the surface of the object fails and a three-dimensional heat flow is generated inside the object. The following temperature changes may occur with or without defects inside the carbon fiber. The first is a scan of healthy material. With the uniform injection of the heat flux, the energy in the healthy material diffuses uniformly inward or from the surface. Observe that the temperature distribution on the surface of the material must be uniform. Another is that there is a defect filled with air inside the material. The heat flow of the continuous scanning of the line laser can be hindered at the defect because the defect will not have the same thermal conductivity as the material. As a result, heat builds up inside the material, resulting in localized hot zones with high temperatures on the surface of the object. During the cooling stage, localized regions or “dark spots” with low temperature appear on the surface due to the low energy contained in the defects. It can be seen in the above situation that due to the different thermal conductivity properties of different objects, corresponding temperature differences will inevitably be formed between defective areas and the non-defective area of the object. Through finite element analysis of the 3D heat transfer model, the characteristics of the defect were detected in both heating and cooling states. In the case of heating, the defect temperature will build up due to thermal resistance, causing the temperature here to be higher than the ambient temperature. However, due to the influences of the properties of the instrument itself, it has been found through many experiments that the weak signal in the cooling stage is easier to observe and analyze [31].

Figure 10 shows the inspection results for impact damage. We have visualized the coordinates of the defect centroids. It can be seen in Figure 10a that every 3 mm defect could be detected with extremely high accuracy. There is a defect in the last image that was not detected because the defect feature was submerged as it left the heated area and could not be detected properly. Figure 10b shows the detection results of 5 mm impact damage. Due to the relatively large size, it could be directly detected by the visual method. In the laser heating experiment, due to its size, the energy loss was relatively fast, resulting in different temperature values in the same target, and the defect features had different properties. Various reasons cannot be taken into account during image processing, resulting in inaccurate center detection. For detection of defects of the same type at different depths, it was found that the detection efficiency was relatively accurate when the defect slightly left or was about to leave the laser heating source, because the energy of the defect was the largest at this time. There was a sufficient temperature difference to appear in the image, and the numerical difference in the pixels was also adequate.

Figure 11 shows tensile fracture defect detection. It can be seen that defects with different fracture angles caused by external forces were detected well. Due to the particularities of such defects, the damage area must be determined to be large or small. For larger defects, it can be easily detected by human eyes or machine recognition. Another situation is that the damage area is very small (fracture type) and the temperature changes too fast, which makes the characteristic signal too weak and difficult to detect. It is also for the reason mentioned that small fracture defects do not have different thermal properties in the composite, which also becomes an advantage. Result founds that the edge features were already submerged at a certain distance from the heat source. Therefore, the closer the defect is to the heat source, the more likely it is to be detected. The visualized results in figure also confirm the statement that the defect angle also affects the detection. Figure 11c shows the case where the third defect has 2 centroids, which is because the defect features are not obvious and are separated during the inspection process.

Table 3 and Table 4 show the detection rates η of different types of defects and the defect angles of the fractures, respectively. Detection rate η calculation method as below:(8)$$\eta =\frac{D(a-a_{0})}{D_{p}}\times 100\%$$
where $D_{p}$ is the pixel value of the defect length, D is the Euclidean distance, a is the center coordinate of the defect, and $a_{0}$ is the detected coordinate value.

The numbers 1–5 in the table represent the fracture labels in different directions, which are consistent with the regulations in the simulation. The last four columns in Table 3 are arranged according to the value of the disc defect depth, and the value increases from 0.5 to 2 mm. From the data in the table, it can be concluded that the detection rate of simulated impact damage is extremely high, and defects of different diameters can be detected. The maximum error did not exceed half of the defect size (<1), which means that the defect localization works very well. For the detection efficiency of defects, the detection accuracy of 3 mm stayed ahead of 5 mm. Thermal energy exhibits different characteristics during cooling. The energy loss of 5 mm defects was faster, and the characteristics of these defects are more complex and changeable, so they have different temperature properties. Another influencing factor is that the spatial angle between the thermal imager and the template cannot be completely vertical, which will also affect the temperature distribution collected. The larger the surface area of the defect, the easier it is to detect manually, and the error of centroid detection is relatively large. The calculation of the fracture angle is replaced by the angle of the smallest circumscribed rectangle of the defect contour. The table data verifies conclusion that the smaller the angle is, the higher the detection rate of the centroid is. This is consistent with the actual simulation results.

As a recap, simulation results showed that the types, sizes, and depths of defects affect the surface temperature. The temperature value also reflects the nature of the defect. Impact defects have the highest detection rates and the most obvious visual effects. From an experimental point of view, controlling the laser energy will not have much impact on the characteristics and structure of the sample. According to the defect location detection results, the impact damage detection rate is the highest. Line laser scanning can detect defects as small as 3 mm and at least 2 mm deep. Fracture defects have weak features and large positioning errors. Using this method, however, localized detection of the defect area can be achieved, followed by localized laser heating of the area to achieve accurate quantification of defect area, length, and fracture angle. At the same time, the study of defect depth can also be completed accordingly. It is worth mentioning that the method developed can be used also for evaluation of internal defects.

## 5. Conclusions

This paper presented a CFRP surface damage detection method based on a controllable line laser thermal excitation source. We basically completed the mutual verification of the damaged area detection by combining a numerical simulation and experiments. With the numerical simulation’s data, we verified that this method is feasible for defect localization. It was found that the temperature change on the material’s surface is related to the size, depth, and fracture direction of the defect. Utilizing the experimental data of laser scanning, we obtained the defect localization results through the image processing algorithm. The results show that the detection accuracy of impact damage is higher, and verified that tensile fractures in different directions will also affect the detection accuracy. When the defect has just entered the cooling phase, the temperature difference between the defect and the healthy area is the largest, and 100% detection accuracy can be achieved at this particular location.

## Figures and Tables

**Figure 1 micromachines-13-00612-f001:**
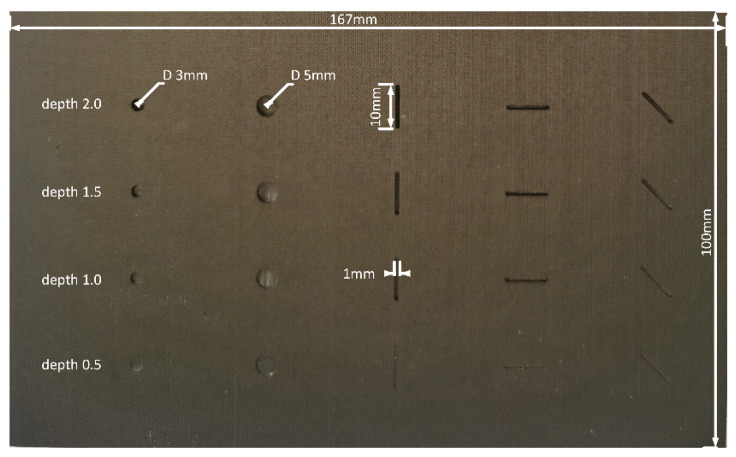
Geometric structure of carbon fiber composite material.

**Figure 2 micromachines-13-00612-f002:**
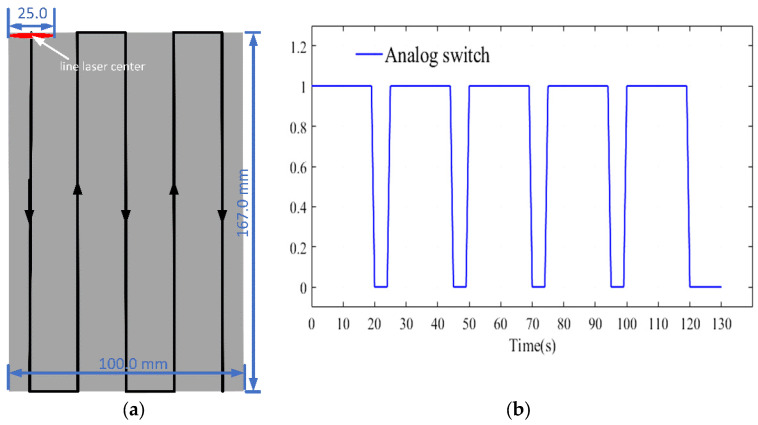
The laser focus moves in a specified way: (**a**) *x*-axis moving mode of laser and (**b**) focus changes along the *z*-axis.

**Figure 3 micromachines-13-00612-f003:**
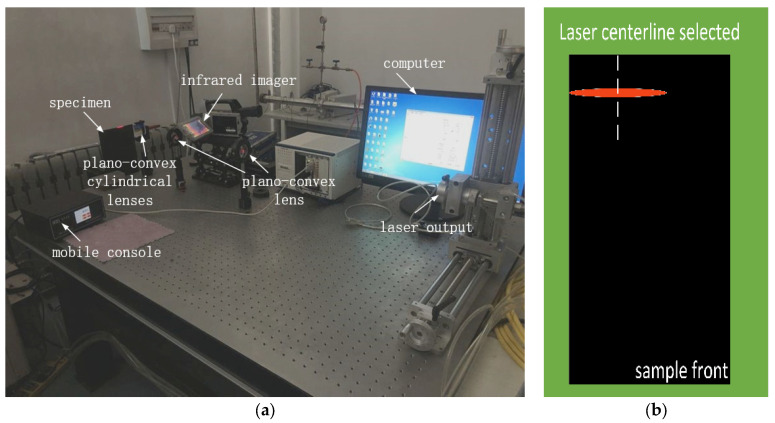
Schematic diagram of the carbon-fiber, pulsed-laser, infrared thermal wave experimental device: (**a**) experimental device diagram; (**b**) the geometry of line-laser.

**Figure 4 micromachines-13-00612-f004:**
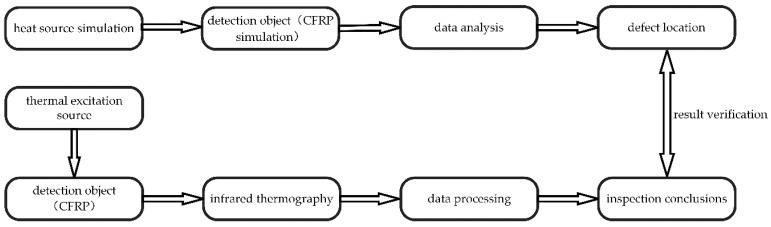
Block diagram of experimental steps.

**Figure 5 micromachines-13-00612-f005:**
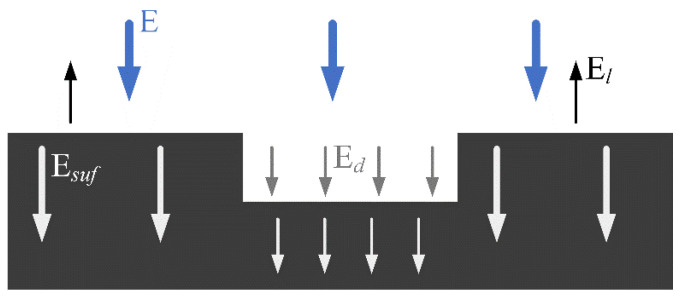
Heat conduction model of carbon-fiber-reinforced composite surface.

**Figure 6 micromachines-13-00612-f006:**
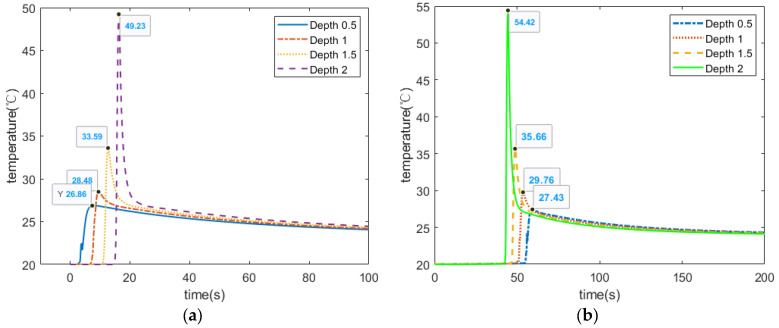
Temperature simulation of impact damage. (**a**) Temperature simulation of 3 mm diameter defects. (**b**) Temperature simulation of 5 mm diameter defects.

**Figure 7 micromachines-13-00612-f007:**
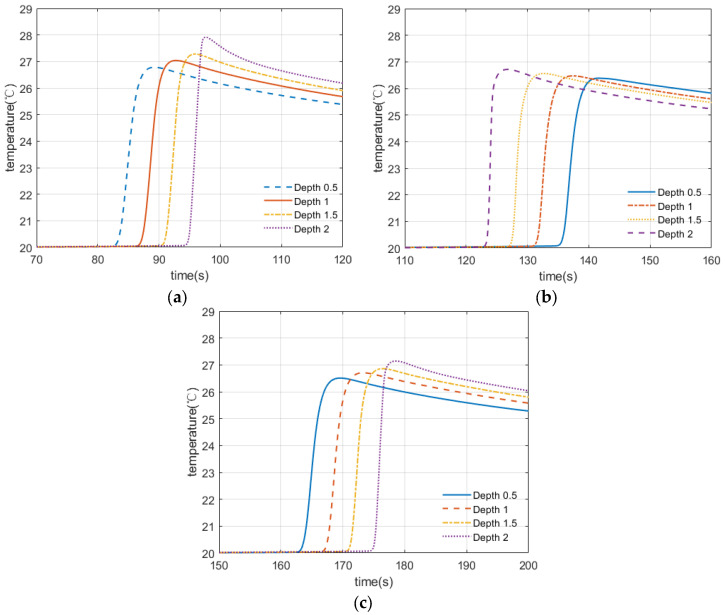
Temperature response of damage in different directions. (**a**) 0° crack temperature response curve, (**b**) 90° crack temperature response curve and (**c**) 45° crack temperature response curve.

**Figure 8 micromachines-13-00612-f008:**
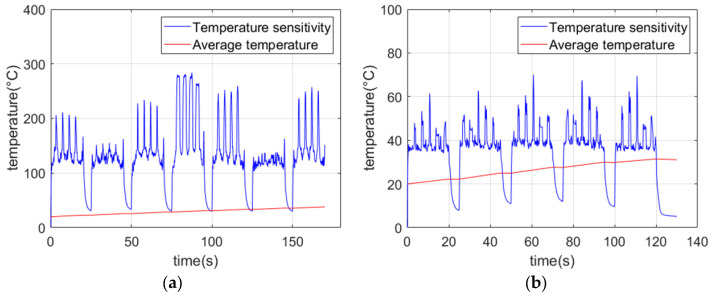
Temperature field versus time curve of carbon fiber surface: (**a**) temperature field changes along the *z*-axis; (**b**) variation in temperature along the *x*-axis.

**Figure 9 micromachines-13-00612-f009:**
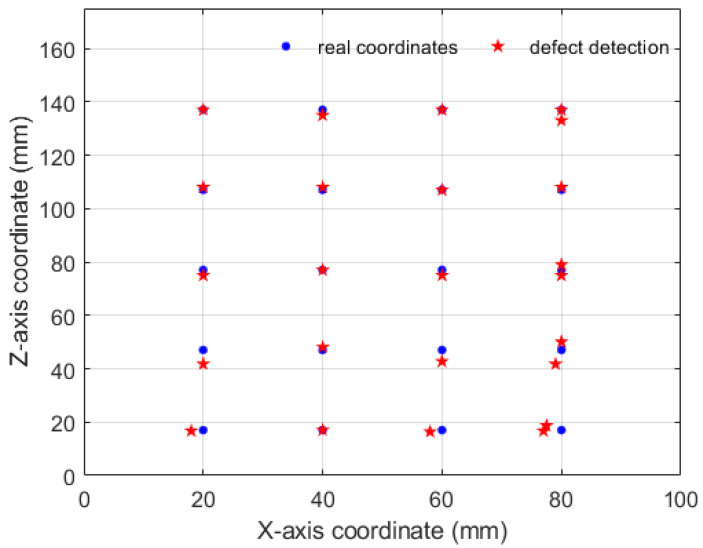
Damage detection results for composite materials via laser scanning.

**Figure 10 micromachines-13-00612-f010:**
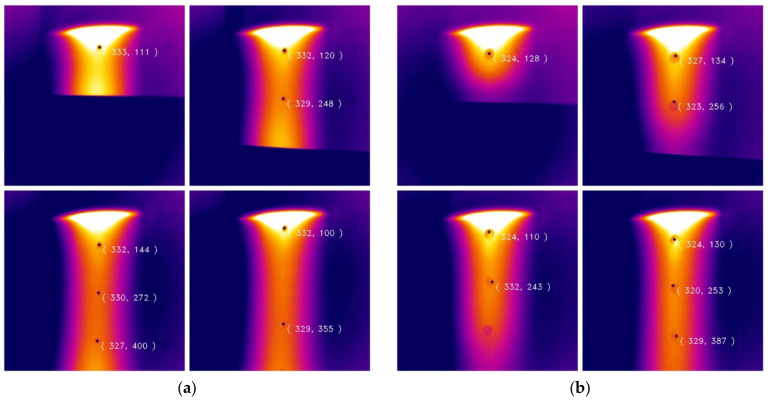
Impact defect detection results for defects of different diameters. (**a**) D3 mm defect detection and (**b**) D5 mm defect detection.

**Figure 11 micromachines-13-00612-f011:**
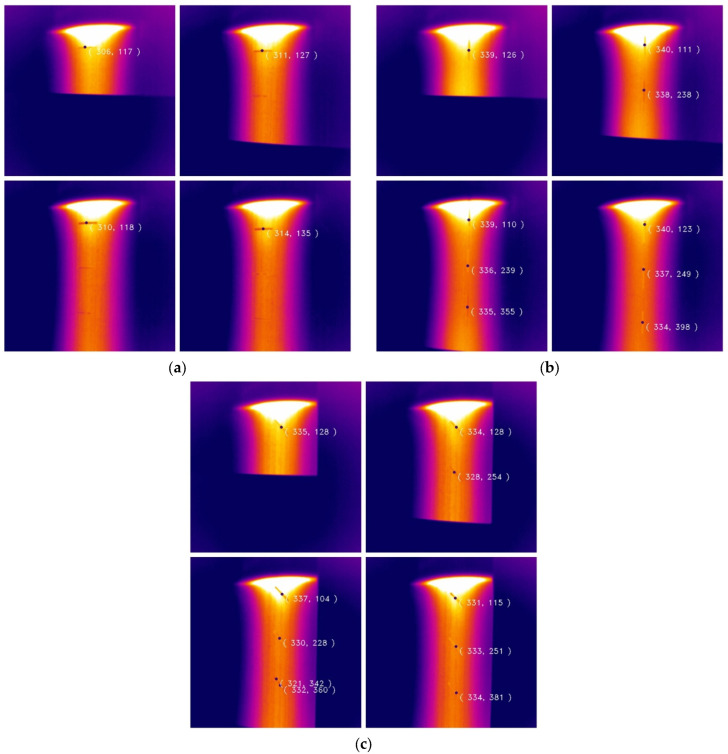
Detection results of cracks in different directions: (**a**) 0° fracture crack detection; (**b**) 90° fracture crack detection; (**c**) 45° fracture crack detection.

**Table 1 micromachines-13-00612-t001:** Thermophysical parameters of carbon-fiber-reinforced composites.

Specimen	Density (g/cm^3^)	Specific Heat Capacity (cal/(g·°C)	Emissivity	Thermal Conductivity (W/(m·K))		
				**x**	**y**	**z**
**C composite**	**1.81**	**0.18**	**0.95**	**5.6**	**0.7**	**0.7**

**Table 2 micromachines-13-00612-t002:** The accuracy of surface damage coordinate detection.

Thickness (mm)	#1	#2	#3	#4	#5
0.5	0.0	0.40	0.40	1.05	0.40
1	1.33	0.40	0	0.22	0.0
1.5	0.0	0.0	0.40	0.85	0.42
2	2.67	0.40	0.40	1.07	0.60

**Table 3 micromachines-13-00612-t003:** Impact defect location detection accuracy.

Size (mm)	$D_{p}\;(\mathrm{Pixel})$	η			
#1	15	0.34	0.20	0.22	0.20
#2	26	0.36	0.25	0.24	0.66

**Table 4 micromachines-13-00612-t004:** Crack defect location detection accuracy and fracture angle.

#3		#4		#5	
$D_{p}=53$		$D_{p}=60$		$D_{p}=54$	
η	Angel(°)	η	Angel(°)	η	Angel(°)
0.15	0	0.07	90	0.08	39
0.04	0	0.00	86	0.04	50
0.20	0	0.24	84	0.13	48
0.12	3	0.7	90	0.34	47

## Data Availability

All data are included in the article.

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
