# Peer review of "Defect Detection Method for CFRP Based on Line Laser Thermography"

_micromachines, 2022, doi:10.3390/mi13040612_

Round 1

Reviewer 1 Report

The article is focused on the development of a method for detecting of surface defects/damage in unidirectional carbon fibre composite by continuous pulse line-laser scanning thermal imaging of the test sample. Even if the manuscript’s topic is interesting, it can’t be published in this form. Several English errors were found and also some phrases are not clear. Expressions like “necessarily necessary”, “temperature response of thermally at the excited surface…” should be avoided.

The “Principle Analysis and Model Structure Introduction” section should be improved as the experiments should be replicate and build by other authors. The materials used should be mentioned: was an unidirectional fabric stitched used? Its characteristics should be mentioned along with the manufacturer or provider, the type of resin along with the curing protocol should be provided. It is mentioned that the lay-up technique was used, it was hand lay-up with a brush or sponge? Or prepreg lay-up?

It is not clear if all values from Table 1 are experimentally determined or were found in the material’s data sheet. It must be mentioned as the characteristics depend on the fibre/resin ratio. Thermal properties of the composite depend on the curing protocol.

The equipment used for manufacturing and studying the samples have to be presented along with the software versions used.

At the bottom of the 5th page is said that “there are no obvious burn marks on the sample’s surface…which indicates that the heat source does not significantly change the thermal properties” It cannot be claimed that there are no significant changes on the thermal properties as the thermal properties were not determined experimentally. If they were experimentally determined, the method used and equipment must be presented. After laser exposure, the polymeric resin can be affected, at most it can be said that there are no physical changes on the laminate’s surface, or there is no physical damage. A thermal analysis could reveal that there are changes in the composite’s thermal properties.

It is recommended to provide images of the experimental setup and manufactured specimens, not just diagrams.

It should be mentioned that the method developed can be used also for evaluation of internal defects. Besides this, in the last sections of the manuscript should be provided information regarding further improvements of the method or future research.

Other template changes should be realised regarding the references format in text and also in the reference section, and also regarding the manuscript font size.  The references and all measurement units should be correctly written, for example reference [21] should be mentioned in text Li et.al. not Y. et al, 13w.

Reviewer 2 Report

Dear Author(s),

The manuscript Micromachines-1672509 and titled “Defect detection method of unidirectional carbon fiber based on line laser thermography” was reviewed. It is good paper.

The research aim is to trial the thermal analysis and image processing to identify the defect of carbon fiber based on non destructive testing principles. Findings from the research were acceptably discussed considering the literature, and conclusion was consistent with generated data. 

It was evaluated that it is interesting paper and finding could attract readers. The manuscript can be accepted after minor revisions.

  1. In Figure 2, please correct “lasre line” as “laser line”.

My best regards,

Round 2

Reviewer 1 Report

Dear authors,

Thank you for updating the manuscript accordingly. The manuscript was significantly improved and it can be published after the editor’s final check.